

# Comparison of mobile and clinical EEG sensors through resting state simultaneous data collection

Ekaterina Kutafina[1,2], Alexander Brenner[1], Yannic Titgemeyer[1], Rainer Surges[3] and Stephan Jonas[4]

[1] Institute of Medical Informatics, Medical Faculty, RWTH Aachen University, Aachen, Germany
[2] Faculty of Applied Mathematics, AGH University of Science and Technology, Krakow, Poland
[3] Department of Epileptology, University Hospital of Bonn, Bonn, Germany
[4] Department of Informatics, Technical University of Munich, Garching, Germany

## ABSTRACT

Development of mobile sensors brings new opportunities to medical research. In particular, mobile electroencephalography (EEG) devices can be potentially used in low cost screening for epilepsy and other neurological and psychiatric disorders. The necessary condition for such applications is thoughtful validation in the specific medical context. As part of validation and quality assurance, we developed a computer-based analysis pipeline, which aims to compare the EEG signal acquired by a mobile EEG device to the one collected by a medically approved clinical-grade EEG device. Both signals are recorded simultaneously during 30 min long sessions in resting state. The data are collected from 22 patients with epileptiform abnormalities in EEG. In order to compare two multichannel EEG signals with differently placed references and electrodes, a novel data processing pipeline is proposed. It allows deriving matching pairs of time series which are suitable for similarity assessment through Pearson correlation. The average correlation of 0.64 is achieved on a test dataset, which can be considered a promising result, taking the positions shift due to the simultaneous electrode placement into account.

Corresponding author
Ekaterina Kutafina,
ekutafina@mi.rwth-aachen.de,
ekutafina@ukaachen.de

## INTRODUCTION

Fast development of mobile and wearable sensors introduces new opportunities for field-trials and long-term monitoring in many research and clinical areas. However, these novel sensors require a rigorous validation, particularly when being used in clinical applications. The methods of validation can differ greatly depending on the type of the sensor and the purpose of the collected data. In this manuscript we focus on the comparison of a novel, wearable electroencephalography (EEG) device to a clinical equivalent with the aim of sensor validation.

In principle, two main approaches to **quantitative** EEG validation can be found in the literature:
**Technical validation** includes an assessment of electrical characteristics of the device or just the electrodes. This approach is often chosen by sensor developers or researchers as an initial validation. For example, *Liao et al. (2011)* report the comparison of dry electrodes impedances to impedances of standard clinical wet electrodes. Generated signals of fixed frequency (*Wyckoff et al., 2015*) can be used for a spectral analysis of the record. Play-back (*Liao et al., 2011*; *Lopez-Gordo, Sanchez-Morillo & Valle, 2014*) is a technique where brain signal is recorded with the gold standard device and then re-played and recorded by the device in question. *In vivo* experiments are challenging, as the signal cannot be recorded at the same time and place with multiple devices. There are two possible solutions (*Lopez-Gordo, Sanchez-Morillo & Valle, 2014*): "same time different place", where several sensors are placed as close to each other as possible and data collected simultaneously; and "same place different time", where the recordings are being performed subsequently at the exact same positions. These methods are also referred to as parallel and serial (*Gargiulo et al., 2010*). In the first case, the recorded signals can be compared through visual analysis (*Gargiulo et al., 2010*; *Estepp et al., 2009*), cross correlation (*Liao et al., 2011*; *Wyckoff et al., 2015*; *Gargiulo et al., 2010*; *Estepp et al., 2009*; *Fiedler et al., 2014*), spectral features (*Grozea, Voinescu & Fazli, 2011*), mutual information (*Quian Quiroga et al., 2002*; *Mikkelsen, Kidmose & Hansen, 2017*) or other numerical features. In case of serial recordings, the options for quality assessment are more limited due to time-dependent changes in brain activity.

Another possibility is a **validation in a given experimental context.** It can be done to answer the question if the new device is suitable for a certain type of research or medical purposes. Here, the choice of possible tests is highly dependent on the application area, but typically the data are collected under well controlled conditions. Both parallel and serial setups can be used.

For instance, the Emotiv Epoc mobile EEG has been investigated by *Badcock et al. (2015)*. The authors compare statistical features of auditory Event Related Potentials (ERP) captured in parallel experiments by the mobile and a research-grade device. *Melnik et al. (2017)* used six different ERP paradigms and serial approach to study the variances caused by multiple sessions, subjects and devices. Another common option is an experiment when the subjects are instructed to have their eyes open and closed for a certain time interval. This allows comparing the EEG alpha band powers (*Wyckoff et al., 2015*).

While all the above approaches are necessary and valid in a research and industrial environment, further validation is required prior to clinical use. Clinical validation proves that a device is capable of performing similar or better than an existing clinical gold standard under real-world conditions. Unfortunately, all described methods require performing the data collection under carefully controlled conditions, which is not always possible in a clinical environment. Some earlier publications (*Titgemeyer et al., 2019*; *McKenzie et al., 2017*) proposed a solution based on professional human assessment, however this approach brings a number of problems, such as subjective judgement and high variation of the scores (*Titgemeyer et al., 2019*). Therefore, an objective, computer-based way of comparison is needed.

The present work aims to answer the following research question:

| Table 1 | Demographic data of the participants. | |
|---|---|---|
| | **Count** | **%** |
| Sex | | |
| Male | 11 | 50% |
| Female | 11 | 50% |
| Age | | |
| <30 | 6 | 27,2% |
| 30–40 | 5 | 22,7% |
| 40–50 | 5 | 22,7% |
| >50 | 6 | 27,2% |

How to compare two multichannel EEG system using a fully computer-based data analysis pipeline and simultaneously recorded data under the following limitations:

- No hardware modification can be done. In particular, the referencing electrodes are placed according to the respective design of the devices.
- No special external stimuli can be introduced, which limits statistical analysis of the signal responses.

The developed pipeline is investigated using simultaneously acquired data from a mobile and clinical EEG during a clinical study at the RWTH Aachen University Hospital.

## METHODS

### Study design

Resting state EEG signals were collected simultaneously with a clinical and mobile EEG devices from a gender-balanced group of 22 patients admitted to the hospital for epilepsy diagnostics. Average age of the patients was 40.2 $\pm$ 15 years and the more detailed demographics can be found in Table 1. All patients were fluent in German language and signed the informed consent. The study was approved by the Ethical Board of Uniklinik RWTH Aachen (EK150-17 3.7.2017) and registered prospectively (DRKS-ID: DRKS00012424). Written consent was obtained from each participant. The primary goal of the study was to assess the quality of the mobile EEG device for epileptiform abnormality detection and the results of the human-based evaluation can be found in *Titgemeyer et al. (2019)*.

The clinical grade data were collected with Brain Quick Plus Evolution by Micromed, which is currently used as a standard device at Epileptology Section at Uniklinik RWTH (Aachen, Germany) where the recordings were conducted. This model has 21 EEG electrodes, ground electrode G1 (positioned between Fz and Cz on the left side) and reference electrode G2 (positioned between Fz and Cz on the right side).

The electrodes are placed according to the 10–20 system (*Tatum, 2013*) (see also Fig. 1A). The data are sampled at 256 Hz. 0.18 Hz high-pass filtering is performed in the amplifier. During patients' hospital stay they were undergoing continuous EEG monitoring for several days, with simultaneous collection of video data. Therefore we use the abbreviation vEEG for the clinical EEG further on. The video data were not used in our study. Approximately
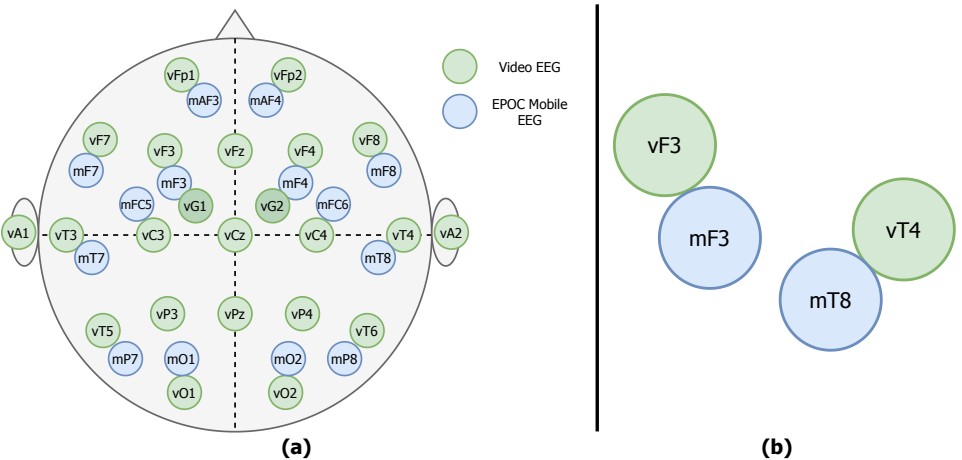

**Figure 1  Example of the vEEG (green) and mEEG (blue) electrode placement.** (A) The placements differ slightly from patient to patient, therefore a sketch is made for each session to track the relative electrode positioning. (B) Zoom to an example electrode quadruple.

every 3 h of EEG records are saved in a separate file and exported in the European Data Format (.EDF files).

For the mobile data collection, the Epoc Emotiv device was used (mEEG). This device was originally marketed as a mobile brain-computer interface (BCI) sensor, but gained a lot of interest from the research community. Previous works showed that this particular device is able to capture EEG potentials and is promising in research context (*Badcock et al., 2015*; *Melnik et al., 2017*). Despite a somewhat limited coverage of the cerebral cortex, the price range and number of electrodes (14, CMS/DRL references) made it a potentially useful tool for clinicians and researchers. The electrode positions are fixed due to a rigid plastic frame and approximate the 10–20 system (see Fig. 1A). Sampling rate of the device used in the study is 128 Hz.

In the experiment Epoc Emotiv was mounted while the clinical EEG device was already in use. mEEG was recorded for approximately 30 min within a single recording session. During these 30 min the patients were asked to limit their movements, as Emotiv Epoc device is highly prone to movement artifacts. At the beginning of the mobile data collection, the participants were asked to blink strongly several times in a row. This was done to facilitate the later data alignment.

The electrodes were placed as close as possible to the vEEG electrodes, at the same time minding the distance enforced by the glue used to attach vEEG electrodes.

The stiff frame of mEEG presented an additional challenge. After the mounting of mEEG a sketch of the placement was made (see Fig. 1A for an example).

Figure 2 shows an illustrative fragment of the same EEG fragment, recorded on both video (a) and mobile (b) devices. Epileptiform abnormalities (spike-slow waves) are visible in both versions.

Before any further processing, the data were pseudonymized and only the session's number was used to keep track of the files.
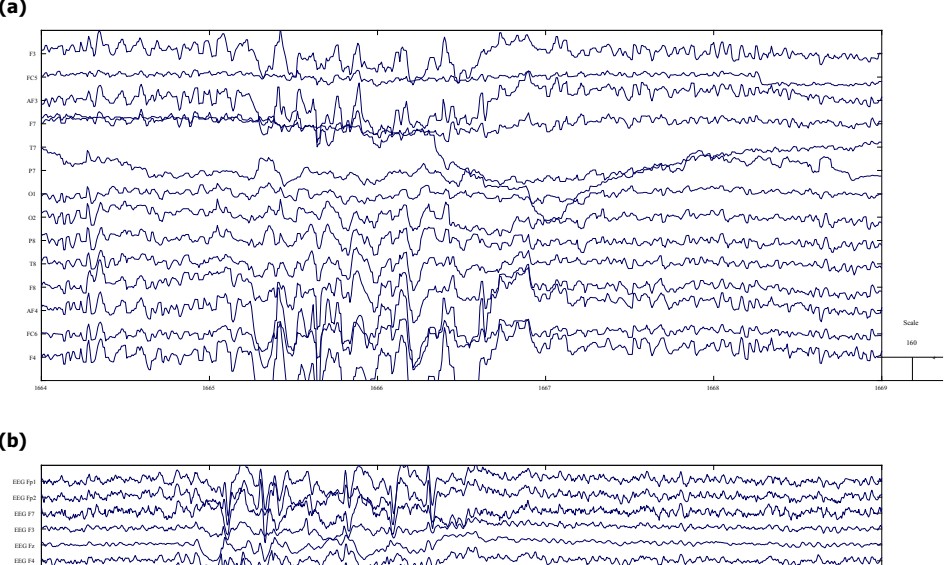

**Figure 2** **Fragment of EEG record with spike-slow wave abnormalities.** (A) vEEG. (B) mEEG.

Three patients' data were discarded from the trial. One set because of the mobile recording software failure, one because of the failure of reference electrode in the clinical EEG, and the last one because of cell phone usage by the patient resulting in strong artifacts.

## Software
Data collection was performed with BrainLab (*Fink et al., 2017*). Data analysis was done with MATLAB R2017b (The Mathworks, Natick, MA, USA) and EEGLAB v.14 (*Delorme & Makeig, 2004*).

## Data preprocessing
The raw EEG data files have different length, sampling rates and electrode placements and, therefore, require several preparation steps before the comparison between mobile and clinical signals can be done. The pipeline presented below allows for constructing aligned pairs of signal vectors, ready for further statistical analysis.

## Initial pruning of clinical EEG data
The duration of the vEEG and mEEG signal is 3 h and 30 min respectively. Based on the blinking and the marked time of the mEEG data collection, approximately 30 min of vEEG

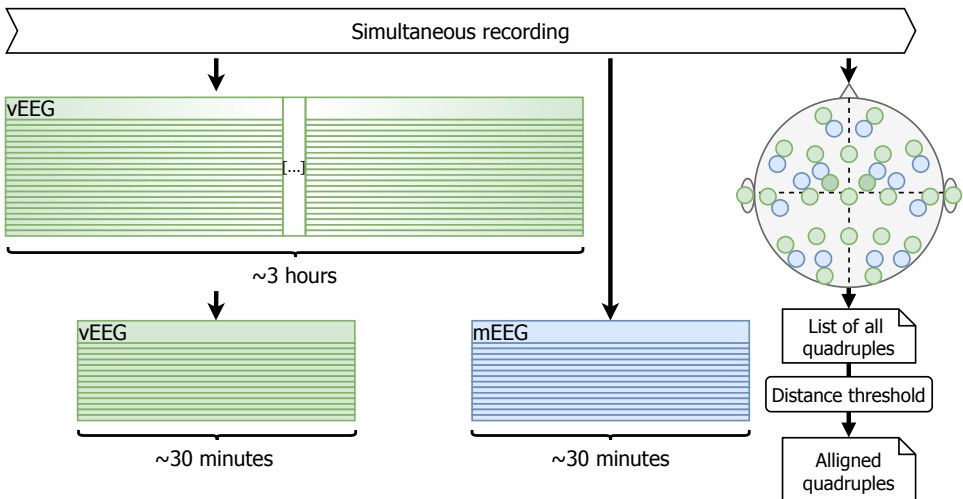

**Figure 3 Initial pruning and quadruple list creation.** Using timestamps and blinking artifacts approximately simultaneous vEEG and mEEG files are obtained. Personal sketch of electrode positions allows to detect spatially close electrode pairs and generate reference-free quadruples.

is cut (see Fig. 3). It is not yet perfectly aligned with the mEEG data, but the shift is within 15 s.

## Selection of corresponding electrodes between vEEG and mEEG

The two considered EEG systems are differently referenced. Therefore, in order to be able to compare the data, we propose a shift to bipolar referencing, defined individually for each patient based on the specific relative electrode positions (see Fig. 1A). The couples of mobile and video electrodes placed directly next to each other (as close as the glue circle allows) are listed. For example, the following couples are chosen (Fig. 1B): mF3-vF3 and mT8-vT4. The set of two corresponding pairs of electrodes will be called a **quadruple**. The list of all possible electrode quadruples (e.g., [mF3, mT8, vF3, vT4], with m* and v* being electrodes from the mEEG and vEEG respectively) is created individually for each patient (Fig. 3). Some quadruples will later correspond to a pair of EEG vectors which we expect to present similarities.

## Choose well-aligned data vectors

The initial choice of quadruples does not guarantee the spatial alignment. In order to assure such alignment, only the pairs which lie far enough from each other will be considered. For example, if we take a signal described by mFC5-mF3 difference and vC3-vF3 (Fig. 1), we can see that the two vectors are far from being well aligned and it is a direct consequence of the spatial proximity of two matched pairs. In order to avoid arbitrary decisions regarding the good or bad alignment we have introduced a decision rule as follows.

(a)  The video electrodes are put on a grid (see Fig. 4, the sides of the squares are assumed to have unit length) and then the Manhattan distance between each pair of electrodes is computed. The distances can take values from zero (from a given electrode to itself), up to six. Electrodes positioned from four to six steps from each other are considered

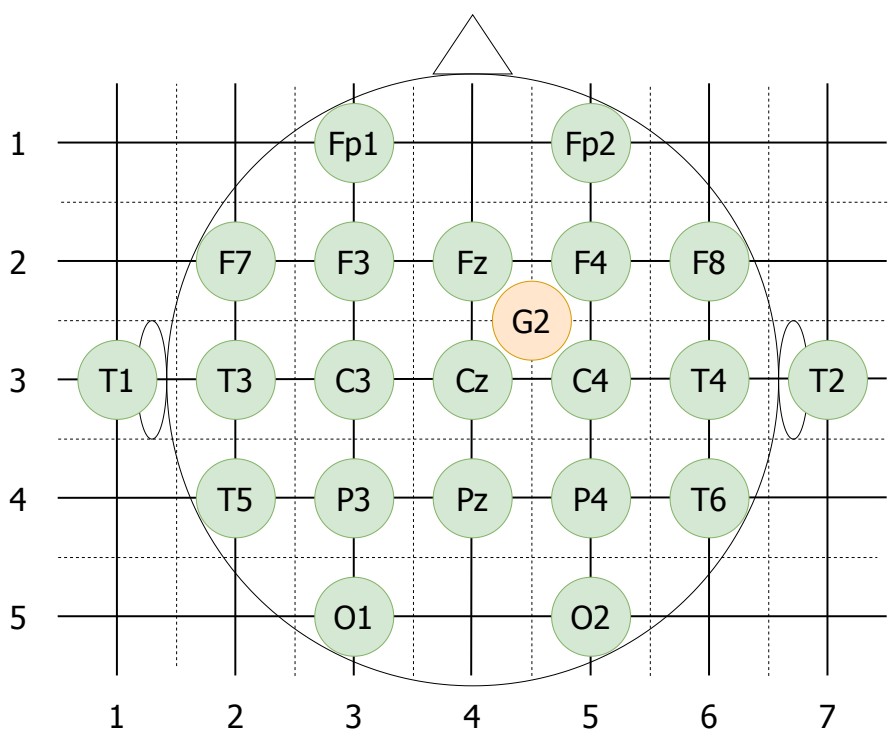

**Figure 4 vEEG electrodes put on an approximate grid.** The grid allows computation of Manhattan distances between different electrodes. Reference electrode G2 is placed in the middle of the unit square to reflect the realistic placement.

to be sufficiently distant. The reference electrode is assumed to be placed exactly in the middle of the Fz-F4-C4-Cz square, and Manhattan distance from Cz to G2 equal to $0.5 + 0.5 = 1$.

(b) For each participant the quadruples with video electrodes separated by Manhattan distance of 4 and higher are taken. Ultimately, 361 quadruples out of 876 possible are chosen.

## Data cleaning and extraction of quadruples

Due to the relatively low signal-to-noise ratio, it is essential to reduce the noise before any comparison is made. Additionally, the data need to be synchronized in time and pruned to the same vector size. Therefore, the following data cleaning pipeline was implemented (see Fig. 5 for the visualisation):

(1) The EEGLAB function *pop_rejchan* is used to detect corrupted channels in all files. The quadruples containing such channels are removed from the list.

(2) Low pass filtering at *fq_lowpass* Hz (the parameter *fq_lowpass* depends on the chosen frequency band).

(3) Downsampling vEEG to a sampling rate of 128 Hz, to match the lower mEEG sampling rate.

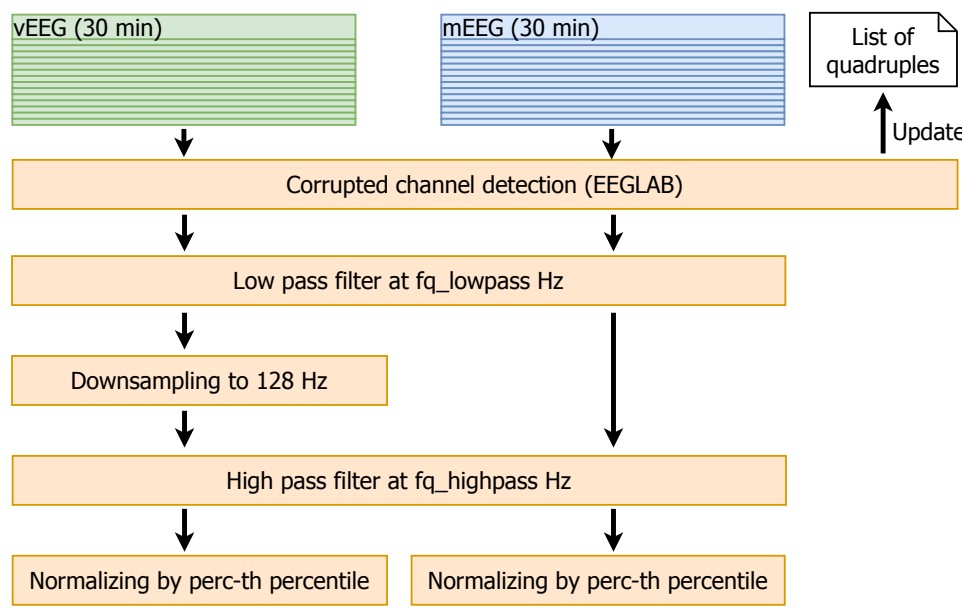

**Figure 5** **Data cleaning.** Data cleaning involves removal of corrupted channels (followed by the update of the quadruple list), filtering, resampling and normalization of the data.

(4) High pass filtering at *fq_highpass* Hz (similarly to *fq_lowpass*, the parameter *fq_highpass* depends on the chosen frequency band).

(5) For each patient, the *perc*-th percentiles of absolute values of the amplitudes are computed for mEEG and vEEG separately, and the corresponding raw data are divided by the resulting value. This normalization allows to avoid the problem of different scales of mEEG and vEEG.

As an output of the described process, we obtain filtered, normalized and roughly time-aligned mobile and clinical EEG data sets for each patient. Additionally the corrupted channels are eliminated and an individual list of channel quadruples to be used for further re-referencing is stored.

## Processing of single quadruples

More precise time alignment and artifact removal is done for individual quadruples (e.g., [mF3, mT8, vF3, vT4] as illustrated on Fig. 1B) of specific patients. The resulting vectors can be compared through Pearson correlation.

## Artifact detection for a single channel

In order to capture short-term signal disturbances, we construct an adjusted procedure for artifact detection:

(1) First, threshold **AmThresh** is fixed and all data with absolute value exceeding AmThresh is marked as NaN (not a number). The data points are not removed to allow later time synchronization. It should be noted, that previous data normalization makes it possible to choose for a common single threshold for both mEEG and vEEG records.

(2) Next, non-overlapping intervals of length **WinLength** are taken, average of amplitudes' absolute values is computed and the whole window is marked as NaN if this average exceeds the parameter **AmThreshWin**.

(3) **Artifact index** is computed by dividing the length of the "corrupted" data (marked as NaN) by the total data vector length.

(4) If **Artifact index** exceeds 70%, all quadruples including this channel are removed from the list.

The described procedure is subsequently applied for all four time series involved in the quadruple.

**Remark 1.** One of the most common procedures for EEG analysis is the removal of eye blinking artifacts. Here, it was decided against this removal, because the locations typically known for strong eye artifacts usually are not involved in the analysis. Additionally, such artifacts are a normal part of the EEG signal, and as such should manifest similarly in mEEG and vEEG signals. One may argue, that high amplitude of such signals may unproportionally influence the linear correlation, but since they are not very prominent in this particular data they were neglected. Similarly, ECG artifacts were not significant in the considered data.

## Re-reference of quadruples

We mathematically re-reference the electrodes within each quadruple (through pairwise subtraction) to eliminate the effect of the global reference (e.g., quadruple [mF3, mT8, vF3, vT4] equivalent to [mF3-mRef, mT8-mRef, vF3-vRef, vT4-vRef] after re-referencing results in a pair [mF3-vF3, mT8-vT4]). This subtraction can be thought of as bipolar (BP) re-referencing. Proximity between corresponding video and mobile electrodes in combination with well-aligned bipolar vectors should result in similarity of those vectors. Therefore, the time series mF3-mT8 is expected to show strong similarities with time series vF3-vT4. Note, that both time series may contain NaN terms propagated from the artifact removal procedure and that the bipolar referencing is different for each patient and is based on patient-specific relative electrode placement. Similarly, the number of quadruples may vary.

## Fine aligning

At this stage, MATLAB function *lag* can be used to find a time shift between the video and mobile data. In order to check how the signal shifts progress in time and capture possible (non)linear drift all signals were divided into 4 equal parts. The lags between mobile and video signals were computed for each quadruple and the resulting median per patient. The results are presented in Table A1, and suggest, that the drift in this particular experiment can be neglected.

What is important to account for, is the fact, that finding the correct lag is only possible if the data has a certain minimal signal-to-noise ratio, otherwise all correlations are close to zero and the time shift is set to a completely wrong value by the algorithm. For one recording session it is possible that the lags slightly vary between quadruples, but typically the differences are within 1–2 data points. Therefore the lags are first computed for all
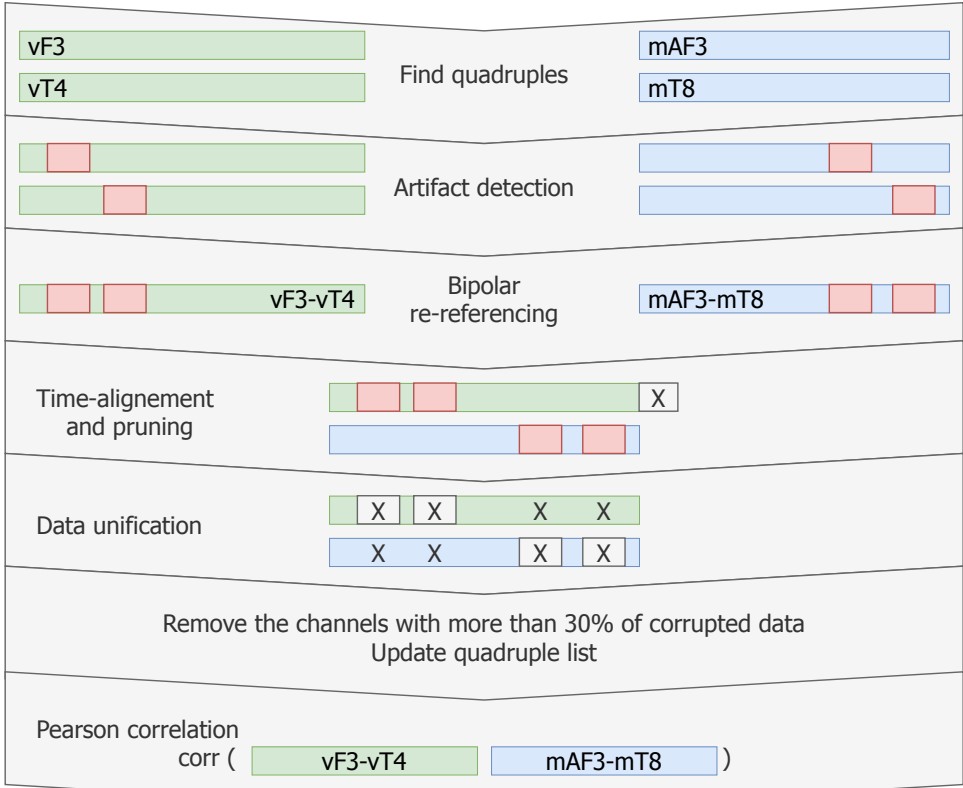

**Figure 6 Pipeline for creating matching data vectors to compare through linear correlation.** For each quadruple artifact detection is performed in each electrode's data, then time aligned reference-free vectors are constructed and the artifacts are removed without breaking the time synchronisation. Vectors with low signal-to-noise ratio are removed from the list. Linear correlation is computed for the remaining matching vectors.

quadruples of a given patient, then the majority vote is used to set the same time shift to all combinations, allowing a fluctuation of ± 5 data points to compensate for differences in device-specific recording order of the channels.

**Remark 2.** If the frequency band chosen for filtering is too narrow, errors might occur in the lag computation. Therefore it is recommended to pre-compute and save lags for a wider band.

Next, the longer vEEG vectors are pruned to match the length of the mEEG vectors and the time stamps from the mEEG data are added. NaN data points from each single time series in the quadruple propagated to the bipolar re-referencing. After the aligning, all data segments where either of the two bipolar time series contained NaN were removed (see Fig. 6).

Finally, Pearson linear correlation for the given quadruple is computed and stored.

## Aggregating data for multiple patients

For each patient and fixed set of the parameters, the table containing all quadruples and their corresponding correlation coefficients is built. The patient average and the grand
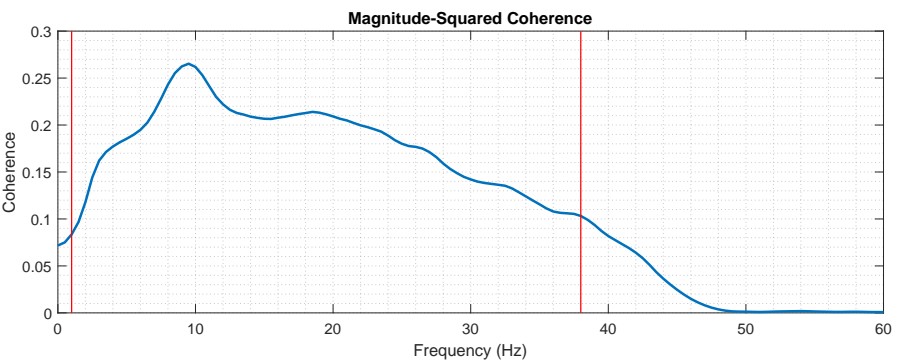

**Figure 7** **Magnitude-squared coherence between mEEG and vEEG averaged across electrodes and patients.** Red lines indicate thresholds for fq_highpass and fq_lowpass.

average across all patients can be reported. Due to the skewed data distribution, Fisher $z$-transform is used to normalize the data before computing the average. Afterwards the inverse transform is performed to return to the original scale. Additionally, the data loss resulting from all processing steps is tracked.

## Optimization and parameters choice

Let us notice that there are 6 parameters which we can choose in the above described procedure: *fq_lowpass, fq_highpass, perc, AmThresh, WinLength* and *AmThreshWin*. A procedure for establishing parameters values consists of several steps. The parameters are optimized for all patients at once resulting in only one set of parameters that is used for all further calculations on all patients. Except for the lag between vEEG and mEEG, no patient-specific parameter is necessary.

First, we fix the default filtering parameters *fq_lowpass and fq_highpass* based on the estimation of the usable spectrum of the data. In principle the frequency parameters can be chosen freely, depending on which part of the spectrum the correlation is of interest. For instance, we will argue later, that alpha band (7.5–12.5 Hz) shows the best correlation. Nevertheless, in order to find the broadest default frequency band reasonable for our research, the average cross-spectrum of the data is computed and thresholding is made to fix the initial fq_lowpass and fq_highpass values (Fig. 7). This cross-spectrum is computed in a procedure similar to the above described correlation computation, with only the initial high-pass filtering of mEEG at 0.5 Hz. Other filtering, normalization and artifact removal are skipped for the estimation of the filter values.

Second, a set of discrete values for *perc, AmThresh, WinLength, AmThreshWin* is chosen. For each parameter combination and patient, the correlation tables are computed.

Third, the data are randomly divided into two subsets of 14 training and 5 test records. These subsets are fixed and no cross-validation is performed. The parameter set with the best resulting grand average correlation coefficient is established based on the first set and quasi-independently evaluated based on the second set.

## Correlations between vEEG electrodes (vEEG to vEEG correlation)

As direct performance comparison, we also investigate the similarities within the vEEG signal and calculate the correlations. In this case, all neighbouring pairs of vEEG electrodes were defined (e.g., [vP4, vT6]), and underwent analogous artifact removal procedures. Instead of quadruples that are re-referenced between two electrodes from both vEEG and mEEG each, we use two vEEG electrodes and their reference electrode. Since both electrodes are referenced to the same reference electrode, this constructs a virtual quadruple (e.g., [G2-vP4, G2-vT6] with G2 being the reference electrode) and correlation between the two signals can be calculated. Similarly to the mobile-to-video case, too "short" quadruples were rejected. Since here the centrally positioned electrode G2 is always involved, the distances may vary from 1 to 4, and the step is now equal to 0.5, as G2 is located between the regular electrodes. 3 was taken as a soft threshold here.

In case of vEEG-mEEG, the difference between the computed correlation and perfect linear correlation value (1) is due to the following factors: (a) devices quality differences, (b) noise, (c) spatial distances between two pairs within one quadruple (e.g., vF3 to mF3 and vT4 to mT8). In case of vEEG-vEEG correlation, the differences are due to (a) noise and (b) spatial distances between the two vEEG electrodes (e.g., vP4 to vT6). These distances are larger than in vEEG to mEEG pairs, but there is no spatial distance influencing the correlation for the reference electrode. If we assume that these differences give a comparable error and the noises are on the same level, then the differences in average correlation should reflect the differences in the devices quality.

An additional step is performed to make the comparison more sensible: since the distances were chosen for mobile-video quadruples [4,5,6] and for video-video quadruples [3,4], we have computed additionally an average correlation for "medium" mobile-video quadruples of the length 3 and 4.

## RESULTS

### Establishing frequency interval for preprocessing

In the first step the frequency band with the boundaries *fq_highpass* and *fq_lowpass* for the preprocessing is chosen for the analysis. The lower boundary (*fq_highpass*) is set to 1 Hz, which is sufficient to remove the drift from the EEG data. In order to set *fq_lowpass*, the magnitude-squared coherence was calculated to assess which frequencies are useful for comparison (Fig. 7). The magnitude-squared coherence indicates the shared information between mEEG and vEEG in distinct frequency bands. A cut-off value of 0.1 was chosen, resulting in upper frequency band limit of 38 Hz. This is in line with the commonly used frequency bands in EEG analysis (following *Zschocke & Kursawe, 2012*, delta up to 3.5 Hz, theta 3.5–7.5 Hz, alpha 7.5–12.5 Hz, beta 12.5–30 Hz and gamma >30 Hz). Recent research also shows the importance of higher gamma (>50 Hz) in certain application contexts (*Ball et al., 2008*; *Darvas et al., 2010*), but the chosen interval of 1–38 Hz is sufficient for the basic neurologic assessment.

**Table 2 Average correlation by frequency band for the test set group.** Gamma is not analyzed individually as only part of the band is present in the signal.

| Average correlation | Delta (1–3.5 Hz) | Theta (3.5–7.5 Hz) | Alpha (7.5–12.5 Hz) | Beta (12.5–30 Hz) | Full band (1–38 Hz) |
|---|---|---|---|---|---|
| Train set | 0.51 | 0.62 | **0.68** | 0.54 | 0.57 |
| Test set | 0.62 | 0.73 | **0.74** | 0.64 | 0.64 |
| Full data set | 0.55 | 0.66 | **0.70** | 0.57 | 0.60 |

## Parameter optimization

For the chosen frequency band (1–38 Hz), we consider the following discrete values of the parameters:

perc = {80%, 85%, 90%, 95%, 97%, 99%},
AmThresh = {1, 1.5, 2, 3, 4, 5, 6, 7, 8, 9, 10},
AmThreshWin = {0.5, 1.0, 1.2, 1.5, 2.0, 2.5},
WinLength = {15, 50, 75, 100, 200, 250}.

For all possible parameter combinations, the average correlation was computed on the randomly chosen train set. The following optimal set of parameters, resulting in mean correlation of 0.57 was obtained through exhaustive search:

perc = 85%,
AmThresh = 8.0,
WinLength = 100,
AmThreshWin = 1.5.

As a boundary condition, the amount of data retained was set to 70% on the train data.

## Correlation between vEEG and mEEG

The optimal parameters determined on the **train** set resulted in a mean correlation of 0.57 and were fixed for evaluation on the test set. When applying the processing chain with the fixed parameters to the **test** set, the resulting mean correlation is 0.64. The volume of data left for the analysis after artifact removal equals 73% of the original data volume.

In order to better understand the distribution of correlations, each patient was analyzed individually (Fig. 8). While most patients produce similar results, some patients have much lower overall correlation coefficients.

The results are highly sensitive to the choice of the frequency band. Considering the fact that the bands of interest might differ depending on the application, the correlations were examined independently for each frequency band (Table 2). The alpha band shows the highest correlation (0.74 on the test set).

## Correlations between vEEG electrodes

In order to obtain the comparison correlation values, the correlation between neighboring video electrodes, positioned at the distance larger or equal to 3 from G2 were computed and averaged on the full data set. The received grand average was 0.78 (see Fig. 9 for details). The corresponding average on full data set for similarly distant (3 to 4) mobile-to-video quadruples was 0.57. To compare the quality of the mEEG and vEEG, we report the
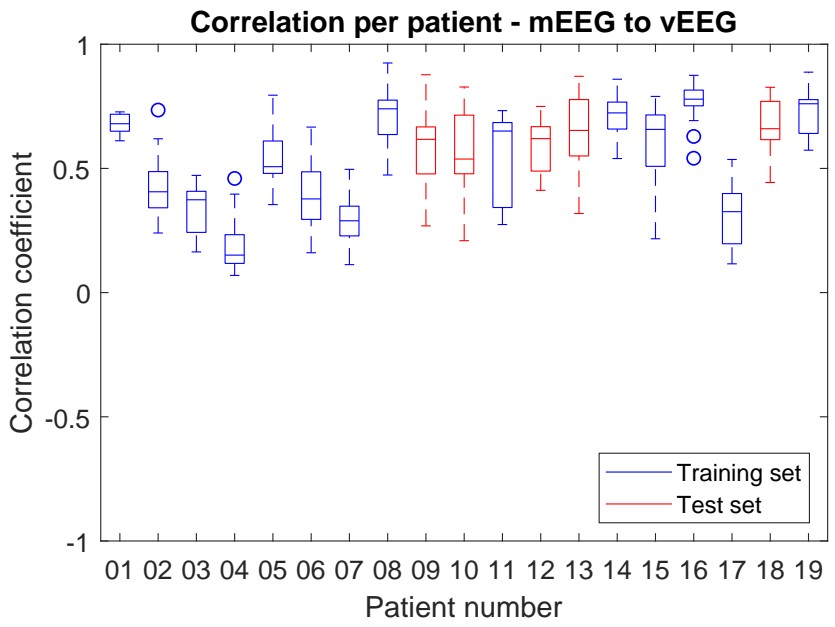

**Figure 8 Distribution of correlations for single quadruples per individual patient.** Results on the test set are shown in red and on the train set in blue.

**Table 3 Percentage of data left after channel rejection and artifact removal.**

| Data left in total (in %) | Test set | Train set | Full data set |
|---|---|---|---|
| mEEG–vEEG ("long distances") | 72.83 | 71.67 | 71.98 |
| mEEG–vEEG ("medium distances") | 73.21 | 73.60 | 73.31 |
| vEEG–vEEG | 95.71 | 89.17 | 90.89 |

percentage of data retained during cleaning for both mEEG to vEEG and vEEG to vEEG correlations (Table 3). The vEEG to vEEG comparison retains approx. 89–96% of the data, which means that 4–11% are discarded. In the mEEG to vEEG comparison, approx. 26–28% of the data are discarded, which indicates that the mEEG is responsible for 15–24% or two thirds of the total discarded data.

# DISCUSSION

## Main findings

The main goal of this work was to find a way to objectively (quantitatively) validate a mobile EEG device in a situation when only resting state brain waves can be collected and no hardware manipulations can be performed (e.g., due to regulations). In particular, the positions of the individual electrodes are fixed and no common referencing is available. To overcome these limitations, we have developed an advanced and multi-parametric data

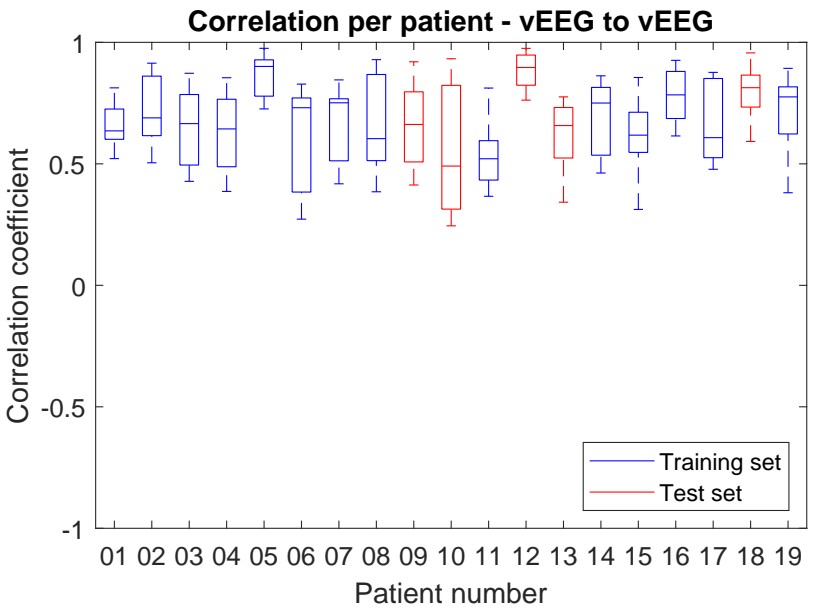

**Figure 9** **Distribution of correlations between pairs of vEEG electrodes over individual patients.**

processing procedure, which allows obtaining illustrative and robust results in the form of a grand average correlation coefficient.

The resulting average correlation is dependent on the chosen frequency range. In case of EEG data it is informative to consider standard brain wave frequency bands (delta, theta, alpha and beta). The average correlation varies from 0.62 on the delta band to 0.74 on alpha band (on the test set). The average across the overall considered frequency band (1–38 Hz) is 0.64. While this number may look moderate, it can be percepted as quite high if we take into account that as much as 73% of all data are preserved and that we compute a grand average over multiple (on average 19 per patient) quadruples of 30 min long data vectors.

For the illustrational purposes we refer to Fig. 10, where two bipolar signals from one quadruple are visibly similar, the quality of the data looks good, but the correlation coefficient is still "only" 0.72.

Due to chance, the average correlation on the test set is higher than on the train set. One of the main reasons is a consistency in signal quality within one record and relatively small ($n = 19$) number of records, resulting in only 5 test records. Nevertheless, both train and test sets show similarity in the results and have similar dynamics across the bands, with alpha band showing the highest correlation.

Although the overall data quality was not optimal, only three out of 22 sessions were rejected because of recording failures, but none during the processing pipeline execution. For the remaining 19 sessions, on average, 19 quadruples of electrodes were selected based on strictly formulated criteria. Furthermore, only 27–28% of data were lost due to rejection of full channels or signal segments (train and test sets). The rejection was performed only based on computer algorithms. Thereby, the reported correlation averages across multiple
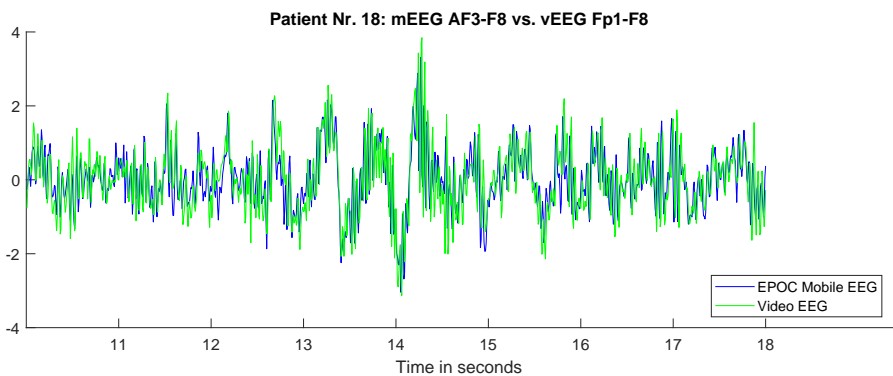

**Figure 10** **Fragment of two bipolar signals from one quadruple.** The signals are processed according to the described pipeline and therefore well aligned. The correlation between the illustrated fragments equals 0.72.

session and quadruples, and covers most of the recorded data. With respect to the high amount of data and potential variation, the low standard deviation of 0.14 (on test set) points towards high robustness.

Due to the uniqueness of the data it is a difficult task to evaluate the results. Therefore we have performed a test to relate the grand average correlation. The average correlation was computed for neighbouring pairs of vEEG electrodes and resulted in a value of 0.78, which lies half way between vEEG-mEEG correlation on "medium" quadruples (0.57) and perfect correlation of 1. The assumptions regarding possible deviations from a perfect correlation listed above are difficult to verify, so we can only hypothesize that these values suggest lower signal quality of mEEG comparing to the clinical device, but the differences seem to be moderate.

## Comparison to the state of the art

Analyzing other works on electrodes quality comparison, where experiments were performed in a more controlled environment, may bring a better understanding of how much of the signal differences are caused by spatial shift of the electrodes and how much by the lower quality of the mobile hardware. Only experiments of "same-time-different-place" type with resting state or similar conditions were chosen for the comparison. In some papers, the authors compare the same type of electrodes in order to estimate the spatial shift-related signal change.

In *Fiedler et al. (2014)* the authors place 3 types of dry electrodes at Fp1, Fp2, O1 and O2 sites, with standard wet electrodes adjacent. Additionally two sets of wet electrodes was tested to provide a baseline. In resting state the resulting average correlations were: 0.24, 0.59, and 0.25 for three dry-to-wet comparisons respectively and 0.58 for wet-to-wet combination.

*Estepp et al. (2009)* under the open eyes condition reported 0.84, 0.61 and 0.32 for dry-to-wet comparison at Fz, C4 and Pz positions respectively. Similarly, wet-to-wet combinations resulted in 0.97, 0.95 and 0.80.

In *Wyckoff et al. (2015)* the measurements with dry and wet electrodes were done at Fz, C3, Cz, C4 and Pz. For the open eyes condition the average correlation varied from 0.28 on delta band to 0.99 on alpha, beta 1 (13–16 Hz) and beta 2 (13–21 Hz) bands.

*Liao et al. (2011)* reported respectively 0.95 and 0.91 correlation at F10 and POz for two different electrode types.

More references can be found in a review paper of *Lopez-Gordo, Sanchez-Morillo & Valle (2014)*. The results in the above referenced papers are characterized by high variability. The reported correlations vary from 0.25 to 0.97 on time-domain signal, which can be explained by different quality of the tested electrodes, but also by differences in the placement, experimental details and data processing. All the experiments were carefully controlled, electrode number limited to a maximum of 5, subjects movement could be minimized and sometimes the segments of data rejected after visual examination (*Gargiulo et al., 2010*).

In contrast, in our research, the data were collected under minimal control, multiple electrode sites were used (including the ones known for high artifact presence) and no human examination was used for data processing. Yet, the results are revealing a correlation level comparable to other comparisons with a similar setup.

## LIMITATIONS

In the described work Pearson correlation was used as a straightforward similarity measurement. It seems to be a reasonable choice in the given context, as linear relationship is exactly what we are expecting from our experiments if the spatial shift is neglected. On the other hand, this spatial shift may introduce significant nonlinear effects. In the further work it might be beneficial to consider different measurements of similarity, such as mutual information (*Mikkelsen, Kidmose & Hansen, 2017*). Nonlinear relationships in EEG have been deeply studied in the context of epileptic seizures where synchronisation of different brain areas often occurs. In *Quian Quiroga et al. (2002)* a number of nonlinear measurements is discussed, however the same paper also suggests ultimate resulting similarity of the different types of measurements, including linear correlation.

Another limitation is the restriction of the comparison bandwidth from 1 to 38 Hz. The maximum available frequency of our setup is 64, however only little mutual information was detectable in the higher frequencies. This indicates both a limitation in the hardware, as well as in the proposed algorithm, as the parts with potentially worse correlation are excluded. While this is reasonable as long as the values chosen are still clinically relevant, it requires special care when choosing the *fq_lowpass* value.

## CONCLUSION

In this work we have developed a data analysis procedure designed to deal with two sources of EEG data recorded simultaneously during resting state. This procedure aims to provide an objective measurement of the data quality.

It is not uncommon that in the procedure of EEG comparison visual assessment by the trained specialists is used as a part of data pre-processing (*Gargiulo et al., 2010*). While

professional opinion may provide a unique insight, it is also very costly to obtain. In the particular case of our study, more than 20 h of multichannel data need to be analyzed. Moreover, multiple studies have shown that human assessment is not fully reproducible and high intra- and inter-rater variances are consistently reported (*Grant et al., 2014*; *Benbadis et al., 2017*), which could be reproduced on our data in a previous publication (*Titgemeyer et al., 2019*). In contrast, automated analysis pipeline provides an objective, fast and low-cost way to perform the data comparison.

The presented procedure deals with the challenges of different referencing, spatial shifting of the electrodes and lack of controlled stimuli (such as in ERP experiments). Using automatically optimized parameters for the pre-processing, a grand average linear correlation of 0.64 between mobile and clinical EEG devices was obtained. It was compared to several baseline correlations, such as clinical-to-clinical EEG correlation, to conclude that the overall quality of the considered mobile device is good, since similar correlations can be seen when only electrode types are changed (e.g., when comparing wet and dry electrodes). This result agrees with our previous study, where trained neurologists were clinically investigating the data (*Titgemeyer et al., 2019*), and with a number of other studies done on Epoc Emotiv in different contexts (*Melnik et al., 2017*; *McKenzie et al., 2017*). However, to our best knowledge, a fully automated approach in combination with the resting state data was not previously reported.

The presented pipeline might benefit in the future from including more sophisticated signal processing methods, such as mutual information. Nevertheless, in the current form it already shows high efficiency and might be potentially generalizable to different multi-channel sensors, such as EMG or ECG.

## ACKNOWLEDGEMENTS

The authors are grateful to Jonas Schulte-Coerne for valuable discussions on the signal processing and to prof. Peter König for a very insightful revision, which significantly improved the paper quality.

## APPENDIX

Table A1 presents the median time lags between mobile and video bipolar signals. The signals were divided into 4 approximately equal length vectors in order to track the possible drift of the delay in time. The median is chosen to avoid the influence of the outliers, which are present due to bad quality of the data in some channels. In this case the lags computed based on autocorrelation are not informative.

**Table A1** **Median (computed across quadruples) lag between video and mobile EEG signals.** The signal length is divided into four equal in time parts to track possible drifts. The numbers represent data points (128 Hz sampling rate).

| Patient | Median lag 1st quarter | Median lag 2nd quarter | Median lag 3rd quarter | Median lag 4th quarter |
|---|---|---|---|---|
| 1 | 739 | 739 | 739 | 739 |
| 2 | 584 | 584 | 584 | 583 |
| 3 | 587 | 587 | 614 | 614 |
| 4 | 780 | 780 | 780 | 780 |
| 5 | 1,046 | 1,045 | 1,045 | 1,045 |
| 6 | 493 | 492 | 492 | 492 |
| 7 | 1,136 | 1,136 | 1,136 | 1,135 |
| 9 | 1,185 | 1,185 | 1,185 | 1,176 |
| 10 | 1,106 | 1,106 | 1,106 | 1,105 |
| 11 | 1,286 | 1,286 | 1,286 | 1,286 |
| 13 | 1,218 | 1,218 | 1,218 | 1,218 |
| 14 | 1,240 | 1,240 | 1,239 | 1,239 |
| 15 | 1,260 | 1,260 | 1,260 | 1,259 |
| 16 | 1,245 | 1,245 | 1,245 | 1,245 |
| 17 | 1,275 | 1,275 | 1,275 | 1,275 |
| 18 | 1,404 | 1,404 | 1,403 | 1,403 |
| 19 | 1,330 | 1,331 | 1,330 | 1,330 |
| 20 | 1,350 | 1,350 | 1,350 | 1,350 |
| 21 | 1,252 | 1,252 | 1,252 | 1,252 |

### Funding

This work was supported by the Faculty of Applied Mathematics AGH UST statutory tasks within subsidy of Ministry of Science and Higher Education (Ekaterina Kutafina). There was no additional external funding received for this study. The funders had no role in study design, data collection and analysis, decision to publish, or preparation of the manuscript.

### Grant Disclosures

The following grant information was disclosed by the authors:
Faculty of Applied Mathematics AGH UST statutory tasks within subsidy of Ministry of Science and Higher Education (Ekaterina Kutafina).

### Competing Interests

The authors declare there are no competing interests.

## Author Contributions

- Ekaterina Kutafina and Stephan Jonas conceived and designed the experiments, analyzed the data, prepared figures and/or tables, authored or reviewed drafts of the paper, and approved the final draft.
- Alexander Brenner analyzed the data, prepared figures and/or tables, and approved the final draft.
- Yannic Titgemeyer performed the experiments, analyzed the data, prepared figures and/or tables, and approved the final draft.
- Rainer Surges conceived and designed the experiments, authored or reviewed drafts of the paper, and approved the final draft.

## Human Ethics

The following information was supplied relating to ethical approvals (i.e., approving body and any reference numbers):

The University Hospital RWTH Aachen granted Ethical approval to carry out the study (EK150-17 3.7.2017).

## Data Availability

The raw data was peer-reviewed but cannot be published per the hospital's privacy policy as it contains personally identifiable data. The code is available at Bitbucket (https://bitbucket.org/abrenner95/meeg-videoeeg-pipeline/src/master/).

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
