# Peer review of "Comparison of mobile and clinical EEG sensors through resting state simultaneous data collection"

_PeerJ, doi:10.7717/peerj.8969_

## Round 0.1 · original submission · Major Revisions

Dear Authors,

Please proceed to do major revisions to your manuscript as per comments from the three peer reviewers.

Reviewer 1 ·

Basic reporting

This study aimed to fill the methodological gap and provide a data analysis pipeline designed to compare the simultaneously recorded data from two multichannel EEG devices. This topic is interesting and the paper is clearly written. This paper may be of great interest to the readers of this journal. However, some comments and suggestions are given.

Experimental design

1. They have to show the representative waveforms of both EEG systems and the detailed demographics of the patients.
2. They didn’t show any feasible and plausible reason why they selected Emotiv for this study.

Validity of the findings

None

Additional comments

None

Reviewer 2 ·

Basic reporting

The article submitted present a pipeline for evaluating non-usual EEGs in relation to stablished well known EEGs. The authors make clear the importance of comparing EEG types and how it is usually done in regarding to technical and research validity. The authors point out that there are still limited methods for assessing clinical validity of EEGs. But it is very important to cite some previous attempts to conduct clinical validation of EEG even though they are not ideal (e.g.: TITGEMEYER et al., 2019; BAROUMAND et al., 2018 and LEITINGER et al., 2016) and discuss their limitation. If the methodological gap mentioned on line 102 refers to methods of clinical validation of EEG, than it deserves a second look since a quick search shows that there is literature on this metter available; but if the methodological gap refers to something else it would be important to explicitly state what the methodological gap is.

Beyond that, although the authors clearly stated that the goal of the study was to provide a pipeline for data analysis, it was not clear which questions the authors were trying to answer. The research question could be, for instance: (1) is the mobile eeg as good as the regular one? Or (2) does this pipeline lead us to the same conclusions as other analysis methods? We can only judge a method in terms of the question to be answered. A clear question, and possibly hypothesis about them, would be very helpful.

References
BAROUMAND, A. G. et al. Automated EEG source imaging: A retrospective, blinded clinical validation study. Clinical Neurophysiology, [s. l.], v. 129, n. 11, p. 2403–2410, 2018.
LEITINGER, M. et al. Diagnostic accuracy of the Salzburg EEG criteria for non-convulsive status epilepticus: a retrospective study. The Lancet Neurology, [s. l.], v. 15, n. 10, p. 1054–1062, 2016.
TITGEMEYER, Y. et al. Can commercially available wearable EEG devices be used for diagnostic purposes? An explorative pilot study. Epilepsy and Behavior, [s. l.], 2019.

Experimental design

The pipeline presented appears to be a valuable tool, very well described, straightforward, and useful. Only small suggestion could be made to improve it. On line 125, it could make it even more clear to describe if G1 is ground or reference electrode and where is it positioned. A picture of a participant head with both eeg systems on could implement information provided on line 150 in order to help those aiming to replicate the author’s method.
The software was described only on line 301, it could be better placed before explaining the preprocessing and analysis methods.

Validity of the findings

no comment

·

Basic reporting

The writing style is fine.

The literature is well covered and I've only a few additions.

The structure of the manuscript is good, although I'd ask for more complete documentation of results (see below), which will require additional figures.

The manuscript is advertised as a preprocessing pipeline for EEG data. However, the strong point is the simultaneous recording by two systems. Below I criticise several aspects of the preprocessing pipeline, which in its present state is far from being an example to be followed by others. I'd suggest casting the whole story with a focus on the comparison of EEG systems.

Experimental design

In their manuscript Kutafina et al. report on a data set with EEG recordings of 22 patients simultaneously with two different measurement systems. It then focusses mainly on the preprocessing of recorded data and comparison, i.e. correlation, of the data obtained by the two systems. This is a valuable data set. Further, as patients screening for epilepsy is involved, a direct comparison of detectability of signatures of epilepsy makes the article highly relevant.

Validity of the findings

Well, after the praise above, here are some serious concerns. Below, you find a list labelled according to severity. I shied away from asking additional data (although the time synchronization is in its present form not satisfactory). But it will require a serious effort with reanalysis of data as well as additional analysis. Please thoroughly address all critical issues and the majority of major concerns.

Additional comments

major - The synchronization of recording devices is based on a few hand-waving measures (eye blinking, line 144) and searching for maximal correlation. State of the art is using precisely timed trigger signals send to both recording devices. Using professional tools like lab-stream-layer is highly recommended. Further, the manuscript assumes that after initial synchronization the clocks stay perfectly synchronized (line 242). This is typically not the case. We observe significant linear drift on a regular basis.
Synchronization during recording can not be added without recording fresh data. Ok. Please search for an optimal match of the two recorded data streams at different points during recording and check linear interpolation of a drift. Further, add a section in the discussion covering this issue.

minor - The reasoning of quadruples becomes clear only later in the manuscript. Explain it the first time when you mention quadruples (line 179).

major - As you are using quadruples that are closely spaced, the bipolar vectors are short and the difference between vectors of the two EEG systems might be significant. You can check quadruples that contain two pairs, that are at a larger distance, each containing one electrode of each system. This way you can reduce the variance introduced into analysis by diverging vectors.

minor - The choice of parameters is complex and contains many fudge factors. Please try to justify the choice of parameters as much as possible by principled arguments and trim down the number of free parameters to adjust. Why has this to be done on a per-patient/subject basis? Isn't it possible to fix these parameters once and for all? Is the optimal choice for each patient very different from others?

unclear - in line 280 you perform cross-validation. "Third, the data are randomly divided into two subsets of 14 training and 5 test records. The parameter set with the best resulting grand average correlation coefficient is established based on the first set and quasi-independently evaluated based on the second set." Is this evaluation on the second set reported here or is there another selection process? Then you would need nested cross-validation.

major - line 298 "If we assume that these differences give a comparable error and the noises are on the same level, " this assumption might be not vlaid. Check for better placed quadruples so that the vectors are well aligned. At least, discuss this critically in the discussion.

major - The observed correlation of 46% appears rather low. (it is the maximal correlation on the training set then validated on the test set?? Otherwise it might be even lower.). It gives an explained variance of 0.46^2= 0.21. That is, only 21% of the variance in one recording is explained by the other. This appear a rather low match. Later you state a slightly higher number of 53%, this difference is not clear to me.

minor - did you use Fischer z transformation forth-average-back while averaging correlation coefficients? As they are not Gaussian distributed, this is a must.

---

## Round 0.2 · accepted · Accept

Good news in the pandemic Covid19 situation, your manuscript has been accepted and will undergo processing.

Reviewer 2 ·

Basic reporting

Paper present proper writing regarding language use, reporting style and necessary information covered.

Experimental design

Research aim and methods are adequate.

Validity of the findings

Authors provided appropriate information on data collected, clearly stated.

Additional comments

It is possible to oberved that the authors considered the suggestions made by the reviewers and impproved their mannuscript. Their scientific work and report present the main characteristics that allow it to be shared with the community.

·

Basic reporting

N/A

Experimental design

N/A

Validity of the findings

N/A

Additional comments

I congratulate the authors for this very nice and thorough revision, resulting in a much-improved manuscript. I endorse the publication of this article. Signed, Peter König